

# Level of anxiety and depression among healthcare workers in Saudi Arabia during the COVID-19 pandemic

Abbas Shamsan[1], Mohammed Alhajji[2], Yasmine Alabbasi[3], Ali Rabaan[4], Saad Alhumaid[5], Mansour Awad[6] and Abbas Al Mutair[7,8,9,10]

[1] Research Center, Dr. Sulaiman Alhabib Medical Group, Riyadh, Saudi Arabia

[2] Ministry of Health, Al-ahsa, Saudi Arabia

[3] Department of Maternity and Child Health Nursing, College of Nursing, Princess Nourah bint Abdulrahman University, Riyadh, Saudi Arabia

[4] Molecular Diagnostic Laboratory, Johns Hopkins Aramco Healthcare, Dhahran, Saudi Arabia

[5] Administration of Pharmaceutical Care, Alahsa Health Cluster, Ministry of Health, Alahsa, Saudi Arabia

[6] Commitment Administration, General Directorate of Health Affairs, Medina, Saudi Arabia

[7] Research Center, Almoosa Specialist Hospital, Al-ahsa, Saudi Arabia

[8] Department of Medical-Surgical Nursing, College of Nursing, Princess Nourah Bint Abdulrahman University, Riyadh, Saudi Arabia

[9] School of Nursing, University of Wollongong, Wollongong, NSW, Australia

[10] Department of Nursing, Prince Sultan College of Health Sciences, Dahran, Saudi Arabia

Corresponding author
Yasmine Alabbasi,
yaalabbasi@pnu.edu.sa

## ABSTRACT

**Background.** The coronavirus disease 2019 (COVID-19) pandemic places a high demand on frontline healthcare workers. Healthcare workers are at high-risk of contracting the virus and are subjected to its consequential emotional and psychological effects. This study aimed to measure the level of depression and anxiety among healthcare workers in Saudi Arabia during the early stages of the COVID-19 pandemic.

**Methods.** This was a cross-sectional study; data were collected from healthcare workers in Saudi Arabia using a survey that included the Zung Self-Rating Depression Scale and the Generalized Anxiety Disorder Scale-7. A total of 326 participants took part in the study by completing and submitting the survey.

**Results.** The vast majority of the participating healthcare workers were Saudi nationals (98.8%) working in a public healthcare facility (89.9%). The results indicated that most of the participants had mild levels of anxiety and depression. A total of 72.5% of the respondents had anxiety, ranging from mild (44.1%) to moderate (16.2%) and severe (12.2%). Moreover, 24.4% of the respondents had depression ranging from mild (21.7%) to moderate (2.1%) and severe (0.6%). The generalized linear models showed that the <30 age group (Beta = 0.556, $p = 0.037$) and the 30–39-year age group (Beta = 0.623, $p = 0.019$) were predicted to have anxiety. The analysis revealed that females were more anxious (Beta = 0.241, $p = 0.005$) than males. Healthcare providers working in primary healthcare centers (Beta = −0.315, $p = 0.008$) and labs (Beta = −0.845. $p = 0.0001$ were predicted to be less anxious than those working in other healthcare facilities. The data analysis showed that participants with good economic status had more depression than the participants in the other economic status groups (Beta = 0.067, $p = 0.003$).

**Conclusion.** This study found that the level of anxiety and depression in healthcare workers was mild. The factors that may contribute to anxiety in healthcare workers

included being female, being younger than 30 or between the ages of 31 and 39, working in a specialized hospital facility, and the number of COVID-19 cases the workers dealt with. Economic status was associated with depression. A longitudinal study design is needed to understand the pattern of anxiety levels among healthcare workers over time during the COVID-19 pandemic.

# INTRODUCTION

Frontline healthcare workers are exposed to workplace-related stressors and hazards and they may experience mental health issues (*Koinis et al., 2015*; *World Health Organization, 2021*). In the healthcare industry, workplace stress is persistent for several reasons, including exposure to infectious diseases leading to illness or death (*Al Mutair et al., 2021b*; *Lee et al., 2007*). The continuously high daily workload may produce high rates of anxiety and depression among healthcare workers (*Chen et al., 2021*). Job burnout among healthcare workers is also predominant due to stress related to administering coronavirus disease 2019 (COVID-19) tests as well as other stressors, such as negative coping style and increased workload, that might be risk factors for anxiety and depression among healthcare providers (*Chen et al., 2021*). There is a growing body of evidence pertaining to high burnout levels among healthcare workers in Saudi Arabia that was present prior to COVID-19 (*Al Mutair et al., 2020*; *Al-Omari et al., 2020*).

Globally, approximately six million newly confirmed cases were reported seven days prior to May 2nd, 2020, with nearly 3 million dying due to complications related to COVID-19 (*World Health Organization, 2020*). Seven days before May 2nd, 2020, the United States, Europe, and South-East Asia had 1,330,513, 1,166,859, and 2,709,582 newly confirmed COVID-19 cases, respectively (*World Health Organization, 2020*). The COVID-19 pandemic has affected the mental health of a wide range of individuals, including healthcare workers (*Czeisler et al., 2020*; *Mental Health America, 2020*; *World Health Organization, 2017*). It has been reported that, during outbreaks of severe acute respiratory syndrome (SARS), clinicians suffered from mental health problems (*Lee et al., 2007*; *McAlonan et al., 2007*). Similarly, the prevalence of mental illnesses was high among healthcare workers during outbreaks of Middle East respiratory syndrome coronavirus (MERS-CoV) (*Al Mutair & Ambani, 2020*; *Salazar De Pablo et al., 2020*).

During the 2020 COVID-19 pandemic in Saudi Arabia, there were 504 private and governmental hospitals with a total of 78,596 beds. Saudi Arabia has 95,336 physicians, 196,701 nurses and midwives, 27,529 pharmacists, and 123,973 allied health personnel, representing 43.9%, 42.9%, 35.2%, and 80.5%, respectively, of the total Saudi healthcare workforce (*Ministry of Health, 2020a*). This is significant because maintaining mental health among healthcare workers is essential (*Al Mutair et al., 2017*) to better control the COVID-19 pandemic. The incidence of COVID-19 has been increasing rapidly in all the provinces and cities in Saudi Arabia. The number of people affected by the disease

had reached 41,000 cases as of May 10, 2020 according to the Saudi Ministry of Health (*Ministry of Health, 2020c*). Some of the reported cases were among healthcare workers. In response to the rapid increase in the number of COVID-19 cases, the government has taken drastic measures to control the spread of the virus, including a lockdown and travel restrictions that were introduced in March, 2020 (*Ministry of Health, 2020b*). The ongoing COVID-19 pandemic, along with the measures to control it, including quarantine requirements, can induce mental health issues, such as anxiety and depression (*Al Mutair et al., 2021b*; *Al Mutair et al., 2021a*). This is especially a concern because the rapid increase in the number of COVID-19 cases has caused healthcare centers and hospitals in Saudi Arabia to extend the number of hours healthcare providers work, placing a significant amount of stress. During the early stage of the pandemic (April to May 2021), information related to the psychological impact of COVID-19 on healthcare workers was scarce in Saudi Arabia. Several studies explored the impact of the COVID-19 pandemic on the level of anxiety or depression of healthcare workers (*Al Mutair et al., 2021b*; *Al Mutair et al., 2021a*; *AlAteeq et al., 2020*; *Rathore et al., 2020*; *Eldaabossi et al., 2022*). However, most studies explored the association between basic socio-demographic characteristics and the level of depression and anxiety among healthcare workers. Other socio-demographic characteristics that were not sufficiently studied include the type of healthcare facility (public or private), working facility, and working area (such as ER, ward or ICU). In addition, to the best of our knowledge, no study explored the association between involvement with testing and treating patients with COVID-19, the severity of the COVID-19 cases healthcare workers dealt with and the level of depression and anxiety among healthcare workers. Therefore, this study aimed to measure the level of depression and anxiety among healthcare workers in Saudi Arabia during the early stage of COVID-19 pandemic, extend knowledge, and include other socio-demographic characteristics. The objective was to ascertain generalized anxiety and depression among healthcare workers amid the ongoing pandemic. Specifically, exploring whether the level of anxiety and depression among healthcare workers is associated with their socio-demographic characteristics, such as gender, age, economic status, type of working facility, their level of involvement with COVID-19 patients (testing and/or treatment), and the number of patients with COVID-19 that the healthcare workers dealt with during the pandemic in Saudi Arabia. This study would further contribute to reforming healthcare policies legislated by healthcare administrators to reduce the impact of the pandemic on the phycological health of healthcare workers.

## MATERIAL AND METHODS

### Study design and settings

An online cross-sectional survey study was conducted to test the research hypothesis. This design was employed to recruit a convenience sample of healthcare professionals who work for public healthcare facilities and the Dr. Sulaiman Al Habib Medical Group, the largest private healthcare provider in Saudi Arabia. That medical group operates eight tertiary private hospitals located in different geographical areas in Saudi Arabia. A convenience sample was considered to be the most appropriate sampling design during the early stage

of the pandemic as it is efficient, simple to implement and the principal investigator is employed in Al Habib medical group.

## Study population

This study included females and males, who were Saudi or non-Saudi healthcare workers working in public or private such as Dr. Sulaiman Al Habib Medical Group. Study selection was based on the following inclusion criteria: (1) healthcare worker aged ≥ 21 years old; (2) living in Saudi Arabia at the time of the study; (3) the healthcare worker was responsible for providing direct patient care in an inpatient or outpatient healthcare setting: and (4) spent at least six months in the current clinical unit.

## Ethical consideration

The researchers obtained ethical approval to conduct the study from the Institutional Review Board at Dr. Sulaiman Al Habib Medical Group (IRB log Number: RC20.03.79). Participation in the study was voluntary and the participants were ensured through the online explanatory letter that the information gathered for the study would be kept confidential and would only be used for the study's purposes. Online informed consent was obtained from all the participants and they were able to download and save the consent.

## Sample

The sample size was calculated using G*Power; considering the sample size parameters, moderate effect size, 5% significance level, and 80% power, it was determined that 300 participants would be a sufficient sample size (*Faul et al., 2007*). A total of 500 healthcare workers were invited to participate in the study, and 326 submitted the completed questionnaires, resulting in a response rate of 65.2% (Fig. 1). All of those who consented to participate in the study, submitted their completed questionnaires. The response rate was computed through determining the IP addresses.

An online link to the anonymous survey was developed through the Qualtrics platform and distributed among healthcare workers in Saudi Arabia. Participants received the online survey through social media channels (WhatsApp, Facebook, Twitter, LinkedIn, Snapchat and Instagram) and emails of staff at Dr. Sulaiman Al Habib Medical Group. The online survey was open for respondents to complete from April 26, 2020 to May 10, 2020. This was done to allow sufficient time for healthcare workers to participate in the study taking into consideration their overwhelming busy schedules during the COVID-19 pandemic.

## Instruments

The Zung Self-Rating Depression Scale (SDS) and Generalized Anxiety Disorder Scale-7 (GAD-7) has previously been used in English to measure depression and anxiety levels among healthcare workers. The tools were in the English language as English is the official language used in all hospitals in Saudi Arabia. In addition, the sample included Arabic and non- Arabic speakers. The GAD-7 is a self-rated scale and consists of 7 items ranked using a Likert scale (*Spitzer et al., 2006*). Previous studies have shown that the GAD-7 scale has adequate reliability and validity (*Rutter & Brown, 2017*). The scale score ranges from 1 to 21, and it is classified as mild (5–9), moderate (10–14), or severe (15–21) (*Spitzer et al.,*

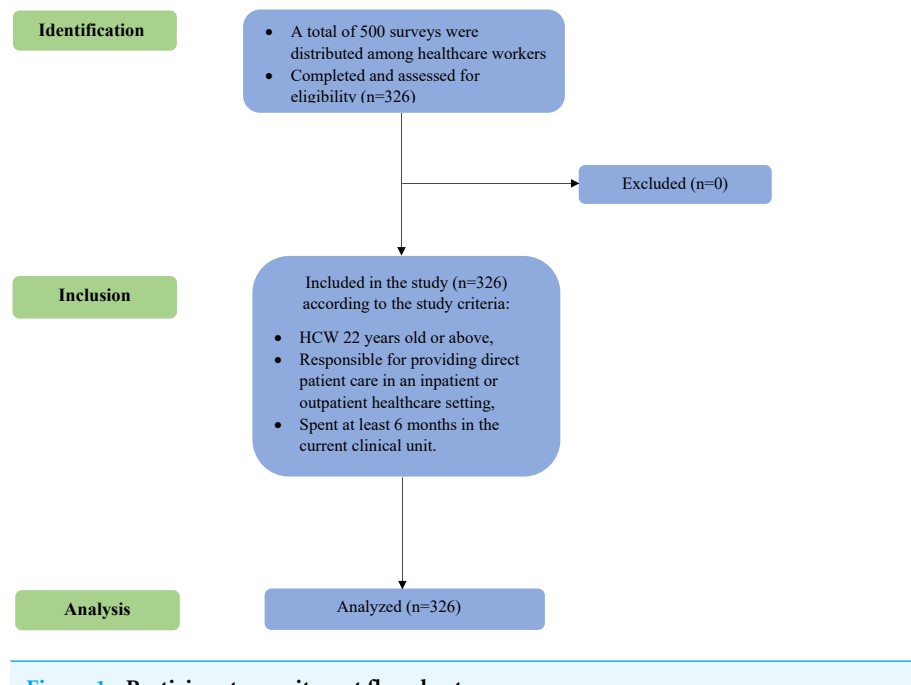

**Figure 1    Participant recruitment flowchart.**

*2006*) based on the total mean score. Scores of ≥ 10 are considered a reasonable cut-point for identifying probable cases of GAD.

The SDS is a tool used to screen depression in different populations. It consists of 20 items ranked on a 4-point Likert scale. The SDS scale score ranges from 20 to 80, with the scores classified as no depression (20 to 49), mild depression (50 to 59), moderate depression (60 to 69), and severe depression (>70) (*Zung, 1967*). The scale has shown adequate validity and reliability with an alpha value of 0.84 (*Dunn & Sacco, 1989*; *Zung, 1967*).

In the current study, the researchers added an additional part to the questionnaire regarding the socio-demographic characteristics of the respondents. These included gender, age, marital status, economic status, nationality, type of healthcare facility, profession, working facility, working area, and number of years of working experience. The socio-demographic part also included whether the respondents had become involved with testing and treating patients with COVID-19 and the level of severity of the COVID-19 cases they dealt with.

## Data analysis

Data were summarized using counts, percentages, medians, and first and third quartiles. Bivariate and multivariable analyses were performed to investigate if the study's two outcomes (anxiety and depression scores) were associated with socio-demographics factors. The distributions of the outcomes were assessed using the Shapiro–Wilk test for normality; it was found that both scores were not normally distributed ($p$-value <0.0001 for anxiety and $p$-value = 0.036 for depression). Therefore, the non-parametric Wilcoxon

rank sum, Mann–Whitney U test, or Kruskal-Wallis was used in the bivariate analyses with the categorical independent variable. The non-parametric Spearman correlations were used for the depression scores and were log-transformed to stabilize the variance, after adding a constant of 1 to each participant's anxiety score due to 0 values on this scale. Then, generalized linear models were used to assess the combined effect of the socio-demographics variables. The best model was found in two steps: first, a full model included the log-transformed outcome and all the predictors. Then, significant predictors ($p$-value <0.05) or those close to significance ($p$-value <0.08) from the full model were included in a reduced model. An F-test of overall significance was used to indicate whether the reduced model provided a better fit to the data then the full model. Because anxiety might be a sign of depression, the anxiety score was included as a covariate in the model with the depression score as the outcome. The level of significance was set at 5%. All analyses were performed using SAS 9.4 statistical software.

## RESULTS

### Socio-demographic characteristics

A total of 326 healthcare workers completed the survey. The socio-demographic characteristics of the participants are shown in Table 1. The majority of the respondents were Saudi Arabian nationals (94.8%), and more than half of the participants were female (58.6%). Half of the respondents were younger than 30 (50.0%) and almost half of respondents were married (49.4%). The economic status of the respondents were: poor (2.1%), good (43.9%), very good (42%), and excellent (12%). Most of the participants worked in a public hospital (89.9%); the remaining (5.2%) worked in the private sector. It was found that 40.5% of the respondents worked in a secondary or a tertiary hospital, 26.7% worked in a specialized hospital, and 20.9% worked in a primary healthcare center. More than half of the participants were physicians (34.7%) and nurses (21.8%) working in different areas of the hospital with an average of $7.35 \pm 6.08$ years of experience. The working areas of the participants included the emergency room (ER) (14.1%), ward (24.2%), intensive care unit (ICU) (7.7%), labs (10.4%), and other areas (43.6%). Almost half (46.3%) of the healthcare workers were not involved with treating or testing patients with COVID-19. The rest of the participants had dealt with, on average, $7.67 \pm 9.78$ patients with COVID-19. The majority of the COVID-19 cases were considered mild (53.1%). Moreover, almost half (46.3%) of all the respondents did not report the level of severity of the COVID-19 cases they dealt with, as they reported having not cared for these patients.

The bivariate analyses revealed that a number of socio-demographic variables were significantly associated with anxiety or depression (Table 1). Gender was significantly associated with anxiety (median score 6.0 (3.0, 9.0) for males and 7.0 (5.0, 11.0) for females, $p = 0.008$) and with depression (median score 39.0 (32.0, 48.0) for males and 43.0 (36.0, 50.0) for females, $p = 0.007$). Economic status was significantly associated with anxiety: (poor median score 16.0 (9.0, 17.0), good 7.0 (5.0, 10.0), very good 46.0 (4.0, 9.0), and excellent 7.0 (3.0, 14.0); $p = 0.006$) and with depression (poor median score 45.0 (40.0,

**Table 1 Socio-demographics characteristics of the participants by anxiety and depressive values (N = 326).**

| Variable | Category | Frequency (%) | Anxiety symptoms Median (Q1, Q3) | P-value Anxiety | Depression symptoms Median (Q1, Q3) | P-value Depression |
|---|---|---|---|---|---|---|
| Gender | Male | 135 (41.4) | 6.0 (3.0, 9.0) | 0.008 | 39.0 (32.0, 48.0) | 0.007 |
| | Female | 191 (58.6) | 7.0 (5.0, 11.0) | | 43.0 (36.0, 50.0) | |
| Age groups | <30 y | 163 (50.0) | 7.0 (4.0, 10.0) | 0.058 | 42.0 (35.0, 49.0) | 0.747 |
| | 30–39 y | 128 (39.2) | 7.5 (5.0, 10.0) | | 43.0 (34.0, 50.0) | |
| | 40–49 y | 27 (8.2) | 6.0 (2.0, 9.0) | | 40.0 (30.0, 50.0) | |
| | >50 y | 8 (2.4) | 4.0 (0.5, 9.0) | | 37.5 (33.5, 45.5) | |
| Marital status | Single | 141 (43.3) | 7.0 (4.0, 9.0) | 0.968 | 40.0 (33.0, 48.0) | 0.181 |
| | Engaged | 5 (1.5) | 7.0 (6.0, 7.0) | | 39.0 (39.0, 50.0) | |
| | Married | 161 (49.4) | 7.0 (4.0, 10.0) | | 42.0 (34.0, 50.0) | |
| | Divorced or Widowed | 19 (5.8) | 7.0 (5.0, 10.0) | | 46.0 (40.0, 51.0) | |
| Economic status | Poor | 7 (2.1) | 16.0 (9.0, 17.0) | 0.006 | 45.0 (40.0, 66.0) | 0.003 |
| | Good | 143 (43.9) | 7.0 (5.0, 10.0) | | 44.0 (37.0, 50.0) | |
| | Very good | 137 (42.0) | 46.0 (4.0, 9.0) | | 39.0 (33.0, 47.0) | |
| | Excellent | 39 (12.0) | 7.0 (3.0, 14.0) | | 40.0 (30.0, 51.0) | |
| Nationality | Saudi | 309 (94.8) | 7.0 (4.0, 10.0) | 0.761 | 42.0 (34.0, 50.0) | 0.280 |
| | Non-Saudi | 17 (5.2) | 8.0 (5.0, 9.0) | | 39.0 (34.0, 47.0) | |
| Type of healthcare facility | Public | 293 (89.9) | 7.0 (4.0, 10.0) | 0.137 | 41.0 (34.0, 49.0) | 0.363 |
| | Private | 33 (5.2) | 8.0 (5.0, 13.0) | | 44.0 (38.0, 49.0) | |
| Profession | Physician | 113 (34.7) | 6.0 (4.0, 9.0) | 0.156 | 40.0 (33.0, 49.0) | 0.606 |
| | Nurse | 71 (21.8) | 8.0 (5.0, 11.0) | | 44.0 (38.0, 49.0) | |
| | Pharmacist | 33 (10.1) | 7.0 (4.0, 10.0) | | 40.0 (36.0, 48.0) | |
| | Working in labs | 33 (10.1) | 6.0 (4.0, 10.0) | | 41.0 (34.0, 48.0) | |
| | Other | 76 (23.3) | 8.0 (5.5, 11.0) | | 41.5 (35.0, 50.0) | |
| Working facility | Primary healthcare center | 68 (20.9) | 6.5 (3.0, 10.5) | 0.023 | 42.5 (34.0, 50.0) | 0.007 |
| | Secondary or tertiary hospital | 132 (40.5) | 7.0 (5.0, 10.0) | | 40.5 (34.0, 48.0) | |
| | Specialized hospital | 87 (26.7) | 7.0 (5.0, 12.0) | | 44.0 (38.0, 51.0) | |
| | Polyclinic | 4 (1.2) | 5.0 (4.0, 4.0) | | 38.0 (33.0, 43.5) | |
| | Lab | 13 (4.0) | 3.0 (1.0, 6.0) | | 34.0 (31.0, 36.0) | |
| | Others | 22 (6.7) | 7.5 (6.0, 10.0) | | 44.0 (39.0, 47.0) | |
| Working area | ER | 46 (14.1) | 7.5 (6.0, 11,0) | 0.313 | 42.5 (36.0, 50.0) | 0.918 |
| | Ward | 79 (24.2) | 7.0 (5.0, 11.0) | | 40.0 (33.0, 49.0) | |
| | ICU | 25 (7.7) | 6.0 (3.0, 9.0) | | 40.0 (36.0, 50.0) | |
| | Labs | 34 (10.4) | 6.0 (4.0, 9.0) | | 40.5 (34.0, 48.0) | |
| | Other | 142 (43.6) | 7.0 (3.0, 10.0) | | 43.0 (35.0, 49.0) | |
| Involvement with COVID-19 | Not involved in COVID-19 cases | 151 (46.3) | 6.0 (4.0, 10.0) | 0.031 | 40.0 (33.0, 48.0) | 0.153 |
| | Diagnosis | 48 (14.7) | 7.0 (4.0, 9.0) | | 42.5 (34.5, 50.0) | |
| | Treatment | 35 (10.7) | 9.0 (5.0, 11.0) | | 43.0 (33.0, 50.0) | |
| | Nursing care | 28 (8.6) | 9.0 (6.5, 15.0) | | 46.0 (43.0, 51.5) | |
| | Other | 64 (19.6) | 7.0 (5.0, 10.0) | | 40.5 (36.0, 47.0) | |
| Level of severity of COVID-19 cases dealt with[*] | Mild | 93 (53.1) | 7.0 (4.0, 10.0) | 0.666 | 43.0 (36.0, 50.0) | 0.928 |
| | Moderate | 47 (26.9) | 9.0 (5.0, 11.0) | | 44.0 (36.0, 48.0) | |
| | Severe | 24 (7.4) | 7.0 (6.0, 10.5) | | 41.5 (37.5, 49.5) | |
| | Fatal cases | 11 (6.3) | 9.0 (4.0, 16.0) | | 37.0 (30.0, 54.0) | |

**Notes.**

*Only includes involvement with COVID-19 cases.

66.0), good 44.0 (37.0, 50.0), very good 39.0 (33.0, 47.0), and excellent 40.0 (30.0, 51.0); $p$ = 0.003). The type of working facility was also associated with anxiety (primary healthcare center median score 6.5 (3.0, 10.5), secondary or tertiary hospital 7.0 (5.0, 10.0), specialized hospital 7.0 (5.0, 12.0), polyclinic 5.0 (4.0, 4.0), lab 3.0 (1.0, 6.0), and other 7.5 (6.0, 10.0), $p$ = 0.023) and with depression symptoms (primary healthcare center 42.5 (34.0, 50.0), secondary or tertiary hospital 40.5 (34.0, 48.0),), specialized hospital 44.0 (38.0, 51.0), polyclinic 38.0 (33.0, 43.5), and other 34.0 (31.0, 36.0), $p$ = 0.007). The anxiety scores of the healthcare workers involved with COVID-19 were higher (not involved in COVID-19 cases, 6.0 (4.0, 10.0), diagnosis 7.0 (4.0, 9.0), treatment 9.0 (5.0, 11.0), nursing care 9.0 (6.5, 15.0), and 7.0 (5.0, 10.0); $p$ = 0.031) in comparison to depression scores (not involved in COVID-19 cases 40.0 (33.0, 48.0), diagnosis 42.5 (34.5, 50.0), treatment 43.0 (33.0, 50.0), nursing care 46.0 (43.0, 51.5), and other 40.5 (36.0, 47.0); $p$ = 0.153). The number of patients with COVID-19 that the healthcare workers dealt with was significantly correlated with the anxiety scores, yet the correlation was weak (correlation coefficient = 0.123, $p$ = 0.026), but was not significantly correlated with the depression scores (correlation coefficient = 0.066, $p$ = 0.232).

## Level of anxiety and depression

The level of anxiety was categorized as mild, moderate, and severe. Scores of $\geq 10$ on GAD-7 needed further evaluation to confirm GAD diagnoses. A total of 234 (72.5%) respondents had anxiety. It was found that 141 (44.1%) of the respondents had mild anxiety, 53 (16.2%) had moderate anxiety, and 40 (12.2%) had severe anxiety (Fig. 2). A total of 80 (24.4%) respondents had depression. Of those respondents, 71 (21.7%) had mild depression, seven (2.1%) had moderate depression, and two (0.6%) had severe depression (Fig. 2).

## Association between the socio-demographic characteristics and anxiety and depression

The results of the generalized linear models were applied to explain the association between the socio-demographic characteristics and the anxiety and depression scores, respectively, as shown in Table 2. In the multivariable analyses with anxiety as the outcome, the F-test of overall significance (F $(21,294)$ =1.04, $p$ = 0.418) indicated that the reduced model, including age ($p$ = 0.005), gender ($p$ = 0.005), type of working facility ($p$ = 0.001), and the number of cases dealt with ($p$ = 0.010), was the best fit for the data. The analysis model suggests that the <30 age group (Beta = 0.556, $p$ = 0.037) and the 30–39-year age group (Beta = 0.623, $p$ = 0.019) were predicted to have significantly more anxiety than the $\geq$ 50 age group. The female respondents were more anxious (Beta = 0.241, $p$ = 0.005) than the male respondents. The analysis also suggests that healthcare providers who worked in a primary healthcare center (Beta = −0.315, $p$ = 0.008) and labs (Beta = −0.845, $p$ = 0.0001 were predicted to have significantly less anxiety than those working in specialized hospitals. Anxiety significantly increased with the growing number of COVID-19 cases (Beta = 0.013, $p$ = 0.009).

   In the multivariable analyses with depression as the outcome, the F-test of overall significance (F$(28,293)$ = 1.14, $p$-value = 0.31) indicated that the reduced model, including

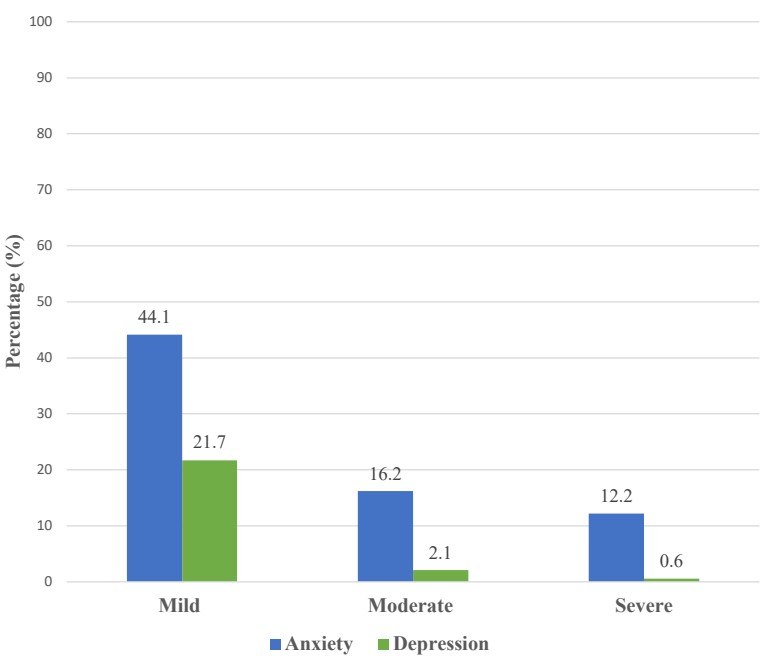

**Figure 2  Levels of anxiety and depression.**

anxiety level ($p < 0.0001$) and socio-economic status ($p = 0.001$), was the best fit for the data. The good socio-economic status group was predicted to be significantly more depressed (Beta = 0.067, $p = 0.003$) than the very good socio-economic status group. The analysis model found that the respondents' anxiety GAD-7 score was significant (Beta = 0.030, $p < 0.001$), indicating that the anxiety level of the respondents predicted significantly higher levels of depression.

## DISCUSSION

The study measured the level of depression and anxiety among healthcare workers in Saudi Arabia during the early stages of the COVID-19 pandemic. Survey results showed that anxiety and depression disorders were prevalent among healthcare workers (72.5% and 24.4%, respectively). Associations were found between some of the demographic characteristics and the level of anxiety. Gender, age, type of working facility, and number of COVID-19 cases dealt with were associated with anxiety, whereas economic status was associated with depression.

The study found that 72.5% of the respondents had anxiety; of these 44.1% had mild anxiety, 16.2% had moderate anxiety, and 12.2% had severe anxiety. A comparable study conducted in July 2020 reported that nearly 50% of healthcare workers experienced anxiety in the Hail region, Saudi Arabia (*Rathore et al., 2020*). *AlAteeq et al. (2020)* reported that 24.9% of healthcare workers had mild levels of anxiety during the early stages of the COVID-19 pandemic in Saudi Arabia and *Al Mutair et al. (2021a)* found that 43.5% of healthcare workers had mild anxiety during that same time period. Consistent with our

**Table 2  Results of the reduced generalized linear model to explain the association between socio-demographics.** Characteristics with anxiety and depression scores.

| Characteristics | N | Beta coefficient (95% CI) for anxiety | Beta coefficient (95% CI) for depression |
|---|---|---|---|
| **Age groups** | | | |
| <30 y | 163 | 0.556 (0.033, 1.079)[*] | NI |
| 30 to 39 y | 128 | 0.623 (0.101, 1.144)[*] | NI |
| 40 to 49 y | 27 | 0.172 (−0.404, 0.747) | NI |
| >50 y (reference) | 8 | 0 | NI |
| **Gender** | | | |
| Female | 191 | 0.241 (0.073, 0.408)[**] | NI |
| Male (reference) | 135 | 0 | NI |
| **Working facility** | | | |
| Primary healthcare center | 68 | −0.315 (−0.545, −0.085)[**] | NI |
| Secondary or tertiary hospital | 132 | −0.099 (−0.297, 0.099) | NI |
| Polyclinic | 4 | −0.288 (−1.017, 0.440) | NI |
| Lab | 13 | −0.845 (−1.270, −0.421)[***] | NI |
| Others | 22 | −0.111 (−0.453, 0.232) | NI |
| Specialized hospital (reference) | 87 | 0 | NI |
| **Number of COVID-19 cases dealt with** | 175 | 0.013 (0.003, 0.024)[**] | NI |
| **Economic status** | | | |
| Poor | 7 | NI | 0.050 (−0.096, 0.195) |
| Good | 143 | NI | 0.067 (0.022, 0.111)[**] |
| Excellent | 39 | NI | −0.055 (−0.122, 0.013) |
| Very good (reference) | 137 | NI | 0 |
| **Anxiety -(GAD-7) score** | | | 0.030 (0.026, 0.034)[***] |

**Notes.**
[*] $P < 0.05$.
[**] $P < 0.01$.
[***] $P < 0.001$.
NI, Not included in the final model.

finding, a study conducted in China found that 44.6% of the respondents had symptoms of anxiety (*Lai et al., 2020*).

The present study found that the female respondents were more anxious than the male respondents and healthcare workers in the <30 age group and those in the 30 to 39 age group were predicted to be significantly more anxious than those in the ≥ 50 age group. During the early stages of the COVID-19 pandemic, a similar study conducted in Saudi Arabia in March 2020 measured depression and anxiety levels among healthcare workers and found that the female participants had a significantly higher mean score for anxiety ($8.11 \pm 6.17$, $p < 0.00$) than their male counterparts (*AlAteeq et al., 2020*). Moreover, participants in the 30–39 age group had higher levels of anxiety than the participants in the other age groups ($7.40 \pm 6.59$, $p < 0.001$) (*AlAteeq et al., 2020*). Another similar finding among Chinese healthcare providers found that female healthcare workers experienced symptoms of anxiety, depression, distress, and insomnia during the COVID-19 pandemic (*Lai et al., 2020*). This may be partially attributed to the fact that females under the age

of 39 also had children and responsibilities at home during the lockdown as school-aged children were restricted from attending school. If females have children, they may tend to have concerns about contracting the virus and transmitting it to their children, which may also increase their anxiety levels.

Compared to healthcare providers working in primary healthcare centers and labs, those working in specialized hospitals were associated with a higher level of anxiety. To the best of our knowledge, no previous studies compared the anxiety levels of healthcare providers working in primary healthcare centers and labs and those working in specialized hospitals during the pandemic. The high level of anxiety might be because healthcare workers in specialized hospital provide services that focus on specific medical needs and care for a population that requires them to use technology and possess a particular subset of skills. In contrast, if healthcare workers provide services in primary healthcare centers, they usually deal with a broad range of health services in the community and refer complex cases to specialized hospitals, thus they may be less anxious than those working in specialized hospitals.

Although the occupation of healthcare workers was not found to be associated with anxiety or depression, the current study found that anxiety among healthcare workers significantly increased as the number of COVID-19 cases they dealt with grew. A similar global study of 75 countries explored anxiety among healthcare workers that dealt with patients with COVID-19 and found that healthcare workers, mainly nurses who dealt directly with COVID-19 cases, were significantly associated with increased levels of anxiety (*Cag et al., 2021*). These findings suggest that the level of anxiety primarily increases if healthcare workers care for patients with COVID-19 in terms of diagnosis, treatment, nursing care, etc. The occupation of healthcare providers in Saudi Arabia should be considered when assessing emotional well-being because workers in some occupations, such as nursing, are more prone to burnout than others, such as physicians and respiratory therapists (*Al-Omari et al., 2020*). This might be due to differences in the working hours between various healthcare specialties.

In terms of economic status, this study found that healthcare workers in the good socio-economic status group were significantly more depressed than those in the very good socio-economic status group. A similar study conducted in Turkey compared healthcare workers' levels of anxiety, depression, and stress between first peak and second peak COVID-19 groups (*Gündoğmuş et al., 2021*). That study found a significant difference in income status between the first peak and second peak COVID-19 groups ($x2 = 52.743$, $df = 2$, $p < 0.001$); moreover, the income status of the healthcare workers deteriorated in the second peak of COVID-19, resulting in an increased level of depression, anxiety, and stress (*Gündoğmuş et al., 2021*). During the COVID-19 pandemic, a Saudi-based study assessed overall emotional well-being and emotional predictors in a Saudi population and found that socio-economic status was one of the predictors of emotional well-being (*Al Mutair, Alhajji & Shamsan, 2021*). The COVID-19 crisis increased the amount of expenditures due to inflation, which may increase the pessimism of individuals (*Binder, 2020*). Therefore, if the socio-economic status was low during the pandemic, it may affect the mental health of healthcare workers.

The current study found that the GAD-7 score was significant, indicating that the level of anxiety of the respondents predicted significantly higher levels of depression. Similarly, a systematic review and meta-analysis of 66 studies that examined the relationship between depression and anxiety found that anxiety symptoms strongly predicted depressive symptoms (*Jacobson & Newman, 2017*). However, the effect size results were small and possibly not clinically significant (*Jacobson & Newman, 2017*).

### Limitations

This study has some limitations. It used a cross-sectional study design, convenience sampling, surveys were distributed online, and it included frontline and non-frontline healthcare workers. Therefore, caution should be taken when generalizing the results to all healthcare workers in Saudi Arabia. In addition, important variables were not examined, such as experience with previous pandemics, as we considered the time to fill out the survey during the high workload of the healthcare workers during the pandemic.

### Implication for practice

This cross-sectional design does not support changes in clinical practice. It is strongly recommended, however, to encourage screening healthcare providers for anxiety and depression at times of pandemics. With special consideration for female healthcare workers, being younger than 30, ranging in age from 30 to 39, working in a specialized hospital, and dealing with a higher number of COVID-19 cases as they were associated with anxiety. In addition, it's encouraged to screen healthcare workers with good socio-economic economic for depression.

### Implication for research

This study has the potential to become a longitudinal study for tracking the prevalence of and factors related to levels of anxiety and depression in healthcare workers. Moreover, a psychological assessment for anxiety and depression should be conducted at another point during the pandemic. Additional studies should be conducted to target frontline healthcare workers in Saudi Arabia with a consideration of supporting those who are not directly caring for COVID-19 patients. In addition, qualitative studies should be conducted on healthcare providers working in primary healthcare centers, labs, and specialized hospitals to explore the impact of COVID-19 on anxiety and depression among healthcare workers.

## CONCLUSIONS

In conclusion, this cross-sectional Saudi based study found that, overall, the generalized anxiety and depression among healthcare providers during the early stage of COVID-19 pandemic was classified as mild. Although anxiety was mild, nearly 70% of the participants experienced it. Factors, such as being female, being younger than 30, ranging in age from 30 to 39, working in a specialized hospital, and dealing with a higher number of COVID-19 cases were associated with anxiety. Economic status was associated with depression. Screening healthcare providers for anxiety and depression at times of pandemics must be considered. More research is needed to understand the pattern of depression and anxiety levels among healthcare workers over time during the COVID-19 pandemic.

## ACKNOWLEDGEMENTS

The research team would like to thank the healthcare workers who completed the survey and participated in the study.

### Funding

The authors received no funding for this work.

### Competing Interests

Abbas Shamsan was employed by the Dr. Sulaiman Al Habib Medical Group. The authors declare there are no competing interests.

### Author Contributions

- Abbas Shamsan conceived and designed the experiments, performed the experiments, analyzed the data, authored or reviewed drafts of the article, and approved the final draft.
- Mohammed Alhajji conceived and designed the experiments, analyzed the data, prepared figures and/or tables, and approved the final draft.
- Yasmine Alabbasi conceived and designed the experiments, performed the experiments, analyzed the data, prepared figures and/or tables, authored or reviewed drafts of the article, and approved the final draft.
- Ali Rabaan performed the experiments, authored or reviewed drafts of the article, and approved the final draft.
- Saad Alhumaid conceived and designed the experiments, performed the experiments, prepared figures and/or tables, and approved the final draft.
- Mansour Awad performed the experiments, prepared figures and/or tables, authored or reviewed drafts of the article, and approved the final draft.
- Abbas Al Mutair conceived and designed the experiments, analyzed the data, authored or reviewed drafts of the article, and approved the final draft.

### Human Ethics

The following information was supplied relating to ethical approvals (i.e., approving body and any reference numbers):

Institutional Review Board at Dr. Sulaiman Al Habib Medical Group (IRB log Number: RC20.03.79).

### Data Availability

The raw data is available in the Supplemental File.

### Supplemental Information

Supplemental information for this article can be found online at http://dx.doi.org/10.7717/peerj.14246#supplemental-information.

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
