# Peer review of "Level of anxiety and depression among healthcare workers in Saudi Arabia during the COVID-19 pandemic"

_PeerJ, doi:10.7717/peerj.14246_

## Round 0.1 · original submission · Major Revisions

I have now received referee reports for your paper entitled "Level of anxiety and depression among healthcare workers in Saudi Arabia during the COVID-19 pandemic”, which are detailed below. As you can see, the reviews suggest that MAJOR REVISION of your paper is required.

Reviewer 1 ·

Basic reporting

Introduction would have been richer if included more worldwide data and info

Experimental design

selection of participants is not clear

Validity of the findings

No comments

Additional comments

The study explores anxiety and depression but no "help" is provided to participants

Annotated reviews are not available for download in order to protect the identity of reviewers who chose to remain anonymous.

·

Basic reporting

'no comment'

Experimental design

The method section is not described sufficiently. It needs revision in relation to the time when the study was conducted. This information is relevant to making statements in relation to the findings. For example, we have information that the study was conducted in the early stage of Covid-19 but we do not have data on which wave of the pandemic this period corresponds to as well as how many cases with Covid-19 had the country during this period.
In addition, more information need to be provided in relation to the study hospital setting, if it treated Covid-19 patients during the study period and how many.
There is no clear situation of informed consent. The authors say that distributed the questionary online while an informed written consent was obtained; how and when is not clear.
Also, the online distribution method is not clear, email, phone, etc, and where that found the data of participants to sent them the questionary.

Validity of the findings

The validity of findings depends a lot on the methodology. In this regard, the authors should revise carefully this section. In addition, there is no clear focus in relation to primary health care workers and hospital healthcare workers. I think that is the novelty of the article and these details should be elaborated more by the authors.
In addition, the conclusions are not stated so well. Also, the limitation section is generalized.

Reviewer 3 ·

Basic reporting

This manuscript describes psychological effects (generalized anxiety and depression) of COVID-19 pandemic among healthcare workers. The manuscript adheres to the journal’s guidelines but the description at several points throughout the paper are lacking which needs to be addressed by the authors. Therefore, I recommend major revisions for this paper.

Experimental design

1. It would be better to divide first section (“Design”) of methods into 2 or 3 subsections e.g. study design and settings, study population, ethical considerations etc.
2. Authors did not mention (neither in abstract nor in the study design section) that it was an online, cross-sectional survey anywhere except the “sample” section. It would be better to mention it in the study design section e.g. “an online cross-sectional study was conducted…” or “A web-based, cross-sectional study was carried out ….”
3. Line # 124-130: At first, it seemed that the investigators used hard-copies questionnaire to collect data because the authors stated they had obtained written informed consents rather than e-consent. However, on line # 138-139, authors stated that they used an online anonymous questionnaire for data collection. Authors need to clarify this issue by detailing the informed consent process as well sampling method in the methods.
4. G*Power should also be bibliographed.
5. Line # 134-136: Authors should report numbers of participants at each stage of their study e.g. total numbers of individuals approached or distributed the questionnaire, consented to participate, included in the study (reasons for exclusion), and included in the final analyses. If possible, add a flowchart of participant recruitment process.
6. Authors stated a response rate of 65.2% computed using the number of potential participants contacted via the convenience sampling as the denominator rather than the total number of potential respondents in the HCW population pool. Therefore, this is more of a completeness rate. To calculate accurate response rates in an e-survey, you must know how you determined a unique user identifier. There are different methods available e.g. IP addresses or cookies or both. Once you know the total unique site visitors, you can calculate view rate (ratio of unique survey visitors/unique site visitors) and participation rate (agreed to participate divided by unique first survey page visitors) and completion rate (ratio of users who finished the survey/users who consented to participate).
7. Please indicate how the multiple entries from the same individual were prevented during the data collection process.
8. Was the study tool/instrument in the English language or Arabic? If it was in English language, provide the rationale for not using an Arabic version of GAD-7 or Zung Self-Rating Depression Scale.
9. For GAD-7, a cutoff score of ≥ 10 should be used as it has a sensitivity of 89% and a specificity of 82%.
10. Provide references for line # 146-147.
11. Kindly include a copy of the survey as a supplementary file.

Validity of the findings

Results:
1. Line # 186: “A total of 326 health professionals responded to the survey”. On the contrary, in the methods section, you mentioned “326 submitted the completed questionnaire”.
2. Line # 191: “Most of the participants worked in a public hospital (89.9%)”. This is very confusing as the authors stated they recruited “a convenience sample of healthcare professionals who work for the Dr. Sulaiman Al Habib Medical Group, the largest private healthcare provider in Saudi Arabia. That medical group operates eight tertiary private hospitals located in different geographical areas in Saudi Arabia”.
3. Similarly, at line 192-193, authors mentioned that 20.9% of their study participants were providing services in primary healthcare settings. However, in methods, they mentioned their study settings to be eight tertiary hospitals (Dr. Sulaiman Al Habib Medical Group). Authors need to clarify the aforementioned issues as it is raising serious concerns over the credibility of this study.
4. Line 228-229: It would be better to use a cutoff score ≥ 10 on GAD-7 to report the prevalence of generalized anxiety in the study sample.
5. Severity of generalized anxiety as well as depression can be depicted on a single figure rather than having two separate figures for anxiety (figure 1) and depression (figure 2).

Discussion:
1. Authors need to describe all the limitations of their study e.g. biases associated with convenience sampling methods, difficulty to prove causality in a cross-sectional analysis, problems associated with self-reported data etc. Authors should also acknowledge that the clinical assessment for the diagnosis of anxiety and depression was not done as per DSM-5 criteria.
2. Delete the word “t” at the end of line 336:
3. Line # 338-341: Better to describe the implications for practice as well.

Conclusion: No need to describe study’s aims and objective in this section. Better to just highlight the key findings and suggest the action required by the policy makers/health regulators to mitigate mental health issues in health professionals.

Additional comments

Abstract:
1. Please indicate that it was an online survey.
2. Please indicate the sampling technique used to gather data in the methods section of abstract.
3. Please include the details of the study duration.

Introduction:
1. As the statistics related to the COVID-19 pandemic were continually changing, and the data that are presented in the second paragraph (line # 93-96) of the Introduction are more than two year old, it may be best to refocus the second paragraph so that there is less of an emphasis on specific values.
2. Line # 105-109: Authors stated that there is a sparsity of information related to the psychological impact of COVID-19 on healthcare workers in Saudi Arabia but proceeded to mention numerous studies that investigated psychological effects of the pandemic among healthcare workers. Since there is plenty of published work from KSA on the similar issue, it would be good to highlight how your study differs from the earlier reports.
3. It would be helpful for the authors to provide a clear rationale for the study, what the findings will add to what is already known about the mental health responses to the pandemics in general, and what the implications of the findings may be for health care professionals in particular.
4. Line # 110-115: As this was a descriptive cross-sectional study, it is not appropriate for testing a hypothesis. The wealth of material obtained in most descriptive studies allows the generation of hypotheses, which can then be tested by analytical or experimental designs. I suggest
5. I suggest re-writing the entire introduction, keeping in view the study’s aims and objective (ascertaining generalized anxiety and depression among healthcare workers amid the ongoing pandemic).

---

## Round 0.2 · Minor Revisions

I have now received referee reports for your revised paper entitled "Level of anxiety and depression among healthcare workers in Saudi Arabia during the COVID-19 pandemic”, which are detailed below. As you can see, the reviews suggest that a MINOR REVISION of your paper is required.

Reviewer 1 ·

Basic reporting

no comment

Experimental design

no comment

Validity of the findings

Inferences made in discussion need to be stated in the "conditional " form

Additional comments

In Fig 2 need to correct the spelling of "moderate"

Reviewer 3 ·

Basic reporting

This manuscript describes the assessment of mental health sequela in HCWs during the COVID-19 outbreak in Saudi Arabia. Commendations again to the authors for taking on mental health issues in HCWs during the COVID-19 pandemic and thanks to the authors for submitting another revision. This revision answered gaps identified in the first review, however, issues still remain and need to be addressed before the manuscript is publishable.

Experimental design

No comment

Validity of the findings

I highlighted some issues during my first review (in the "Additional Comments" slot), however, they were not adequately addressed. Authors are requested to revise the manuscript considering the comments mentioned below.
1. Line # 110-114: Authors stated that there is a sparsity of information related to the psychological impact of COVID-19 on healthcare workers in Saudi Arabia but proceeded to mention numerous studies that investigated psychological effects of the pandemic among healthcare workers. Since there is plenty of published work from KSA on the similar issue, it would be good to highlight how your study differs from the earlier reports.
2. It would be helpful for the authors to provide a clear rationale for the study, what the findings will add to what is already known about the mental health responses to the pandemics in general, and what the implications of the findings may be for health care professionals in particular.
3. Line # 116-120: As this was a descriptive cross-sectional study, it is not appropriate for testing a hypothesis. The wealth of material obtained in most descriptive studies allows the generation of hypotheses, which can then be tested by analytical or experimental designs. I suggest revising this paragraph, keeping in view the study’s aims and objective (ascertaining generalized anxiety and depression among healthcare workers amid the ongoing pandemic).
4. Line 128-129: Please complete the sentence
5. Line 361: Please correct the spelling of “convenience”
6. Line 387-388: No need to describe study’s aims and objective in the Conclusion.
7. I suggest you to prepare figure 1 (Participant recruitment flowchart) according to the STROBE guidelines/statement.

---

## Round 0.3 · accepted · Accept

I am writing to inform you that your manuscript - Level of anxiety and depression among healthcare workers in Saudi Arabia during the COVID-19 pandemic - has been Accepted for publication.